# Disappearance of Timestep Embedding: A Case Study on Neural ODE and Diffusion Models

**Bum Jun Kim**                                                    *bumjun.kim@weblab.t.u-tokyo.ac.jp*
*Graduate School of Engineering, The University of Tokyo*

**Yoshinobu Kawahara**                                            *kawahara@ist.osaka-u.ac.jp*
*Graduate School of Information Science and Technology, The University of Osaka*
*RIKEN Center for Advanced Intelligence Project*

**Sang Woo Kim**                                                       *swkim@postech.edu*
*Department of Electrical Engineering, Pohang University of Science and Technology*

**Reviewed on OpenReview:** *https://openreview.net/forum?id=bpaLYaf6Dp*

## Abstract

Dynamical systems are often time-varying, whose modeling requires a function that evolves with respect to time. Recent studies such as the neural ordinary differential equation proposed a time-dependent neural network, which provides a neural network varying with respect to time. However, we claim that the architectural choice to build a time-dependent neural network significantly affects its time-awareness but still lacks sufficient validation in its current states. In this study, we conduct an in-depth analysis of the architecture of neural ordinary differential equations. Here, we report a vulnerability of vanishing timestep embedding, which disables the time-awareness of a time-dependent neural network. Specifically, we find that the ConcatConv operation, which is widely used in neural ordinary differential equations, causes an additive effect of timestep embedding, which is readily canceled out by the subsequent batch normalization. This vanishing timestep embedding also arises for group normalization and is analyzed thoroughly with respect to the number of channels, groups, and relative variance. Furthermore, we find that this vulnerability can also be observed in diffusion models because they employ a similar architecture that incorporates timestep embedding to discriminate between different timesteps during a diffusion process. Our analysis provides a detailed description of this phenomenon as well as several solutions to address the root cause. Through experiments on neural ordinary differential equations and diffusion models, we observed that ensuring alive time-awareness via proposed solutions boosted their performance, such as classification accuracy, FID, and inception score, which implies that their current implementations lack sufficient time-dependency.

## 1 Introduction

Deep neural networks have demonstrated remarkable potential in a wide variety of fields, including computer vision and natural language processing. Furthermore, recent studies have extended their usage to the modeling of time-varying dynamical systems. A prime example is the neural ordinary differential equation (NODE) (Chen et al., 2018), which employs a time-dependent neural network to model the differential equation of a time-varying dynamical system. The NODE is able to describe a complex dynamical system via a single neural network that varies with respect to time and has been deployed in several fields (Dupont et al., 2019; Yan et al., 2020; Norcliffe et al., 2020; Kidger et al., 2021; Xia et al., 2021).

Another famous use of time-dependent neural networks is in diffusion models (Ho et al., 2020; Song et al., 2021c). Recent studies on generative models cast their task as a diffusion process, whose modeling requires

discrimination between different timesteps throughout the process. For this task, a time-dependent neural network has been employed, which is suitable to describe different phenomena evolving with respect to time.

Despite different contexts for the emergence of NODE and diffusion models, we discover that they share a similar design pattern in architecture: The time-dependent neural network incorporates time-awareness by introducing a module of timestep embedding, whose values vary with respect to time. From this observation, we collectively analyze the two families of models in terms of their architectural configurations, which are subjects of this study.

Albeit the wide usage of NODE and diffusion models for their application, however, we find that no study has yet been conducted to investigate the validity of the current architectural choice. In other words, we claim that the architectural choice to build a time-dependent neural network still lacks sufficient validation. The lack of architectural analysis of these time-dependent neural networks may cause significant problems: For example, if the architecture has a certain vulnerability, it would perform unwanted behavior and provide degraded performance.

This study inspects the time-dependency of NODE and diffusion models. Eventually, we report a vulnerability in their architecture: The timestep component is inherently prone to vanish, which disables time-awareness. This phenomenon was observed for both NODE and diffusion models owing to their similar design patterns. We provide a unified viewpoint on the vanishing timestep embedding of NODE and diffusion models, along with a detailed explanation of their architecture. We also present several solutions that address the root cause of this problem to guarantee sufficient time-dependency. Through practical experiments on NODE and diffusion models, we observed that encouraging a sufficient timestep embedding significantly boosted their performance.

## 2 Vanishing Timestep Embedding

### 2.1 Background: How NODE incorporates time-awareness

The time-dependent neural networks we discuss indicate a family of neural networks that receive both input feature and timestep. In contrast to the common neural networks that provide fixed output without time-dependency, time-dependent neural network provides a time-varying function, which is suitable for modeling different phenomena that vary with respect to time. Their prime examples in the current research community include NODE (Chen et al., 2018) and diffusion models (Ho et al., 2020), which are the main subjects of our study. In this section, we first review NODE, state its problem, and then extend our claim to diffusion models.

The inspiration behind NODE is to understand the depth of a neural network as time and the residual network (He et al., 2016) as a discrete-time function. In NODE interpretation, a feature map evolves with respect to time in a dynamical system to be an output. They propose a continuous-time neural network that is formulated in terms of an ordinary differential equation (ODE) as

$$\frac{d\mathbf{h}(t)}{dt} = f(\mathbf{h}(t), t, \theta). \tag{1}$$

While NODE $f$ is a neural network parameterized with $\theta$, it receives both input feature $\mathbf{h}(t)$ and timestep $t \in [0, T]$, providing a different output depending on the timestep $t$. Applying a suitable ODE solver to $f$ yields an integration $\mathbf{h}(T)$ from the corresponding initial value $\mathbf{h}(0)$. Note that the term "time" in our study differs from the conventional notion of time used in time-series data. In NODE studies, the time unit is something similar to a depth unit in a neural network, not indicating the real notion of time, while time-series data can be used as input for NODE. The behavior of a time-dependent function evolving in $t \in [0, T]$ can be understood as a behavior of data passing through the layers of a neural network, from the first layer to the last layer, *e.g.*, the 101st layer. In other words, the behavior of a NODE block being time-dependent is similar to a residual network having different functions or weights across different layers. Thus, repeating a NODE block with respect to different times is similar to passing a residual network across whole layers, which enables performing a static task such as image classification and generation. If NODE

Table 1: List of notations used in this study.

| Notation | Meaning |
|---|---|
| $f$ | NODE that is modeled as a neural network. |
| $\theta$ | Parameters of NODE. |
| $\mathbf{h}(t)$ | Input feature in NODE. |
| $t$ | Timestep in NODE. Ranges from 0 to $T$. |
| $\mathbf{X}$ | An input feature map to NODE pipeline. |
| $H, W, C$ | Height, width, and the number of channels for the input feature map $\mathbf{X}$. |
| $\mathbf{J}$ | All-ones tensor. All elements are one. |
| $\mathbf{Z}$ | Output of ConcatConv in NODE pipeline. |
| $\mathbf{Z}^k$ | The $k$-th channel of $\mathbf{Z}$. |
| $\mathbf{W}$ | Convolutional kernel used in ConcatConv. |
| $*$ | Convolutional operation. |
| $[\mathbf{X}; t\mathbf{J}]$ | Concatenation of $\mathbf{X}$ and $t\mathbf{J}$ with respect to the channel dimension. |
| $\tilde{\mathbf{W}} = \mathbf{W}_{1..C}$ | Subset of $\mathbf{W}$ from the first to the $C$-th channel. |
| $\mathbf{W}_{C+1}$ | Subset of $\mathbf{W}$ corresponding $(C+1)$-th channel. |
| $\mathbf{v}$ | Output of convolution on an all-ones tensor with the kernel $\mathbf{W}_{C+1}$. |
| $\tilde{\mathbf{v}}_t$ | Timestep embedding. |
| $\mathbf{v}_{i,j}^k$ | An element of $\mathbf{v}$ that corresponds to the $(i, j)$ coordinate and the $k$th channel. |
| $\mathbb{E}[\cdot], \mathrm{Var}[\cdot]$ | Mean and variance computed through BN manner. |
| $\tilde{\mathbf{p}}_t$ | Positional timestep embedding. |
| $\frac{\partial y}{\partial x}$ | Partial derivative of $y$ with respect to $x$. |
| $n_h \times n_w$ | The spatial size of the kernel. |
| $\sigma^2$ | Variance of weight initialization. |
| LN | Layer Normalization. |
| GN | Group Normalization. |

does not have a time dependency, it becomes a fixed-weight RNN with a residual connection. This behavior is further analyzed later.

Here, we describe the NODE for vision tasks. For an image classification task, an input image is processed by an early-stage such as strided convolution to produce an image feature, which acts as the initial value for the ODE, *i.e.*, $\mathbf{h}(0)$. Applying NODE yields $\mathbf{h}(T)$, which corresponds to the output of the continuous-depth neural network. A head, such as a fully connected (FC) layer and softmax, outputs a final score of classification probability. In other words, adopting a NODE means replacing a sequence of residual blocks with a single ODE block while using the remaining layers such as the early-stage and head.

To build a time-dependent neural network $f$, the original NODE study proposed a pipeline of [GN–Act–**ConcatConv**–GN–Act–**ConcatConv**–GN]. The ConcatConv operation—also referred to as Conv2dTime—is key to incorporating time-dependency. Consider an input feature map $\mathbf{X} \in \mathbb{R}^{H \times W \times C}$, where $H$, $W$, and $C$ correspond to the spatial size and the number of channels of the feature map. The ConcatConv operation generates a timestep feature $t\mathbf{J} \in \mathbb{R}^{H \times W \times 1}$ where $\mathbf{J}$ is an all-ones tensor. The timestep feature is concatenated to the input feature map with respect to the channel dimension, yielding $[\mathbf{X}; t\mathbf{J}] \in \mathbb{R}^{H \times W \times (C+1)}$. Now, a vanilla convolutional operation is applied to produce $\mathbf{Z} = \mathbf{W} * [\mathbf{X}; t\mathbf{J}] \in \mathbb{R}^{H \times W \times C}$, where $*$ indicates the convolutional operation and $\mathbf{W}$ indicates the convolutional kernel. The convolution here blends the timestep feature into all channels. With rich nonlinearities such as group normalization (GN) (Wu & He, 2018) and ReLU activation function (Act), the output becomes time-dependent. Indeed, the pipeline with ConcatConv is considered the most standard practice in Neural ODE. Since its introduction, this pipeline along with ConcatConv has been widely adopted in follow-up studies on NODE (Dupont et al., 2019; Yan et al., 2020; Norcliffe et al., 2020; Kidger et al., 2021; Xia et al., 2021). Although these studies do not explicitly state to adopt the pipeline with ConcatConv, we find that it is implicitly used in their source code.

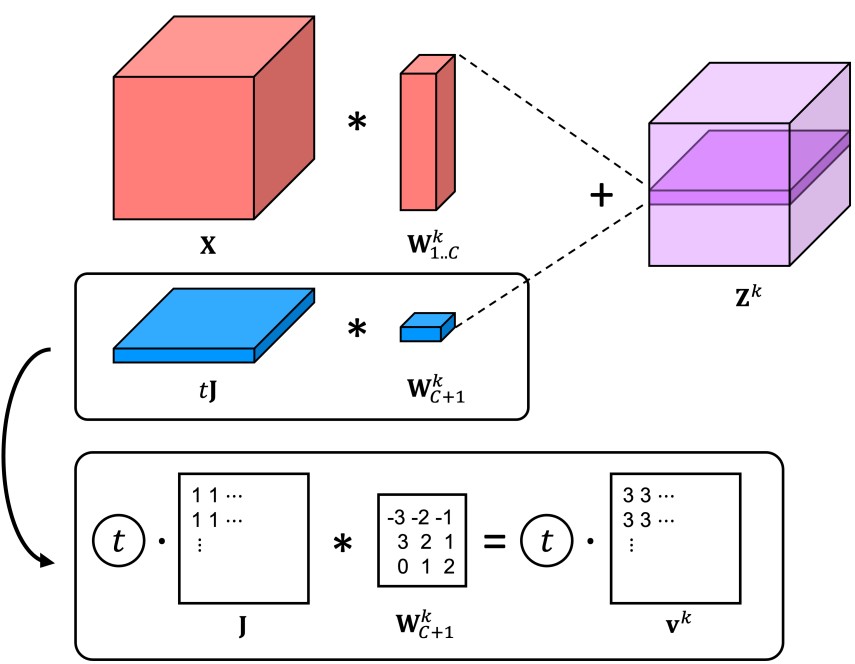

Figure 1: In ConcatConv operation, applying a convolutional kernel $\mathbf{W}_{C+1}^k$ to $t\mathbf{J}$ is equivalent to using $t\mathbf{v}^k$ that has the same element spatially

## 2.2 Problem statement: Timestep embedding is inherently prone to vanish

By definition, the time-dependent neural network should yield different behaviors depending on the timestep. The major claim of this study is that the architectural choice of the time-dependent neural network affects its time-dependency. In other words, certain incorrect architectural designs disable the time-awareness of the time-dependent neural network.

We first show that the NODE-style architecture loses its time-awareness depending on the choice of normalization layer. Here, we revisit the ConcatConv operation. We rewrite the convolutional kernel $\mathbf{W}$ with the last one $\mathbf{W}_{C+1}$ and the rest one $\mathbf{W}_{1..C}$, $i.e.$, $\mathbf{W} = [\mathbf{W}_{1..C}; \mathbf{W}_{C+1}]$. Now, the $k$th channel of the ConcatConv output is

$$\mathbf{Z}^k = [\mathbf{W}_{1..C}^k; \mathbf{W}_{C+1}^k] * [\mathbf{X}; t\mathbf{J}] \tag{2}$$

$$= \mathbf{W}_{1..C}^k * \mathbf{X} + \mathbf{W}_{C+1}^k * t\mathbf{J} \tag{3}$$

$$= \mathbf{W}_{1..C}^k * \mathbf{X} + t\mathbf{v}^k, \tag{4}$$

where $\mathbf{v}^k = \mathbf{W}_{C+1}^k * \mathbf{J} \in \mathbb{R}^{H \times W \times 1}$. Because the last convolution is applied to the all-ones tensor, we obtain $\mathbf{v}_{i,j}^k = \sum \mathbf{W}_{C+1}^k$ for all spatial locations $(i, j)$, which is actually a single parameter that corresponds to the spatial sum of the kernel (Figure 1). This interpretation applies to all $k$th channels. From this observation, the ConcatConv operation is equivalent to 1) applying a vanilla convolution with $\tilde{\mathbf{W}} = \mathbf{W}_{1..C}$ and 2) subsequently applying element-wise addition with a timestep embedding $\tilde{\mathbf{v}}_t = t\mathbf{v}$, which can be written as $\mathbf{Z} = \tilde{\mathbf{W}} * \mathbf{X} + \tilde{\mathbf{v}}_t$. Therefore, the pipeline [GN–Act–**ConcatConv**–GN–Act–**ConcatConv**–GN] is equivalent to a pipeline of [GN–Act–Conv–**Emb**–GN–Act–Conv–**Emb**–GN], where the Emb operation indicates adding timestep embedding. Crucially, for each channel, even though the convolutional kernel is parameterized by nine parameters for a $3 \times 3$ convolution, its output for the all-ones tensor is simply one parameter, which is equivalent to a scalar offset for a feature map.

The core problem we claim is that adding a channel-wise scalar offset is inherently prone to vanish when subsequently applying a normalization layer. Before starting the main topic of GN, here consider batch normalization (BN) (Ioffe & Szegedy, 2015). BN applies channel-wise mean-std normalization, where the

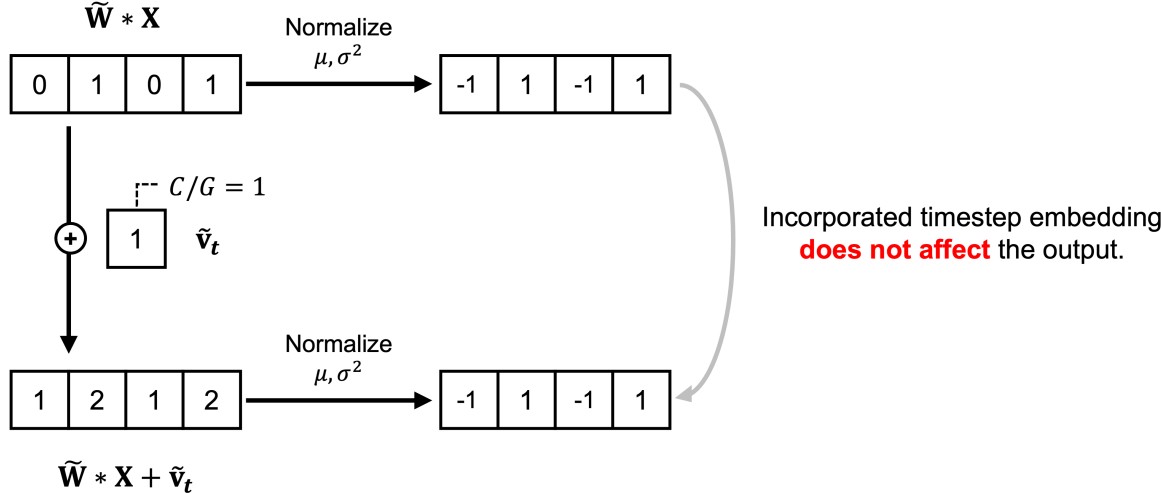

Figure 2: Illustration of vanishing timestep embedding. An additive scalar offset is simply canceled out by the subsequent mean-std normalization.

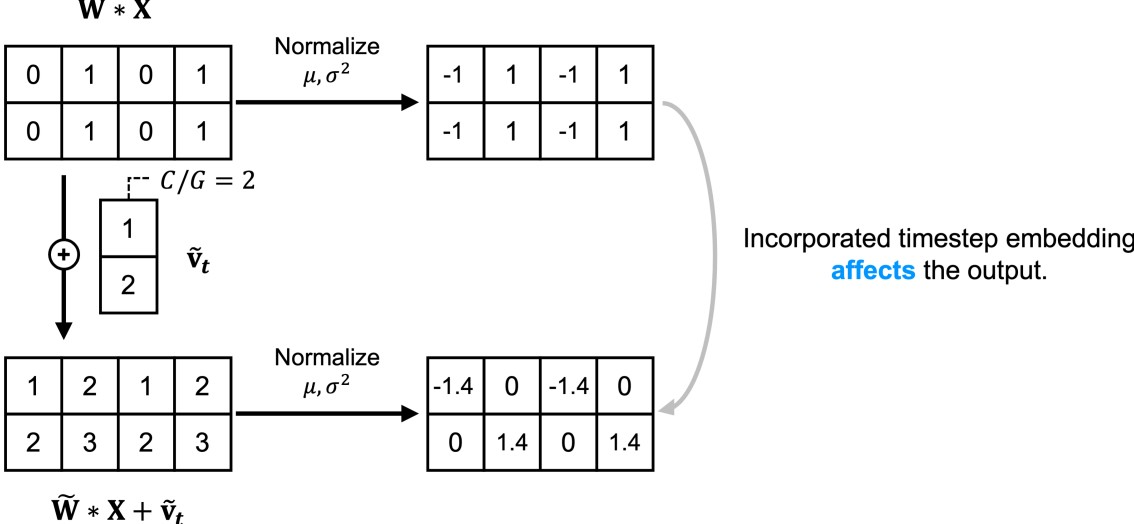

Figure 3: To avoid the use of scalar offset, we should ensure that each normalization unit has several elements of timestep embedding more than one in each channel, which would not be canceled out by the subsequent normalization

channel-wise offset is simply canceled out (Figure 2). Thus, the $k$th channel of the BN output is

$$\frac{(\tilde{\mathbf{W}}^k * \mathbf{X} + \tilde{\mathbf{v}}_t^k) - \mathbb{E}[\tilde{\mathbf{W}}^k * \mathbf{X} + \tilde{\mathbf{v}}_t^k]}{\sqrt{\text{Var}[\tilde{\mathbf{W}}^k * \mathbf{X} + \tilde{\mathbf{v}}_t^k]}} \tag{5}$$

$$= \frac{(\tilde{\mathbf{W}}^k * \mathbf{X}) - \mathbb{E}[\tilde{\mathbf{W}}^k * \mathbf{X}]}{\sqrt{\text{Var}[\tilde{\mathbf{W}}^k * \mathbf{X}]}}. \tag{6}$$

In this example, the timestep embedding $\tilde{\mathbf{v}}_t^k$ cannot affect the output, which disables dependency on the timestep $t$.

In fact, though common neural networks prefer BN, time-dependent neural networks have opted for GN. This practice is in agreement with our analysis: In contrast to BN, which applies channel-wise normalization, GN partitions $C$ channels into $G$ groups, and each group contains $C/G$ channels that are generally larger than one. In this case, the $C/G$ elements affect the mean and standard deviation of the normalization, which would not be canceled out (Figure 3). In summary, to avoid vanishing timestep embedding, we should ensure that each normalization unit has the number of channels larger than one. In this regard, GN and layer normalization (LN) (Ba et al., 2016) are safe to be deployed in time-dependent neural networks because they have $C/G$ and $C$ channels per normalization unit, respectively. However, BN and instance normalization (IN) (Ulyanov et al., 2016) have a single channel in a normalization unit, which disables timestep embedding.

Note that although the existing practice of employing GN with $G = 32$ partially solves this problem, we further claim that their timestep embeddings are inherently prone to vanish owing to this architectural choice and still lack sufficient time-awareness. For example, we later claim that the additive timestep embedding is still prone to vanishing, even with GN, depending on the relative variance, whose fundamental cause lies in the architectural configuration. The objective of our study is to revisit the architectural choice of time-dependent neural networks, providing practical guidelines beyond choosing $G = 32$ to further enrich their time-awareness.

Now we visit the diffusion models. To discriminate between noise levels that have been applied in a diffusion process, diffusion models require time-awareness for each state. The original study (Ho et al., 2020) implemented a denoising diffusion probabilistic model (DDPM) using a pipeline of [GN–Act–Conv–**Emb**–GN–Act–Conv], which is significantly similar to the pipeline of the NODE-style architecture. Since its introduction in DDPM, this pipeline has been widely employed in other generative models, such as NCSN++ (Song et al., 2021c), DDIM (Song et al., 2021a), VDM (Kingma et al., 2021), and Improved DDPM (Nichol & Dhariwal, 2021) (See the Appendix). Similar to the NODE-style pipeline, because normalization is applied after additive timestep embedding, we claim that the DDPM-style pipeline is similarly prone to vanishing timestep embedding. Notably, the timestep embedding in diffusion models is obtained by applying sinusoidal timestep embedding to a multilayer perceptron (MLP) (Ho et al., 2020; Vaswani et al., 2017; Mildenhall et al., 2022), which is different from the linear-time NODE-style timestep embedding $\tilde{\mathbf{v}}_t = t\mathbf{v}$. Although the sinusoidal timestep embedding in diffusion models may seem advanced, they still have only a single element in each channel, which is vulnerable to being canceled out by the subsequent normalization layer. Thus, we claim that the vanishing timestep embedding can also be observed in diffusion models, which should be prevented in advance. This analysis explains why GN has been preferred in diffusion models.

Note that diffusion models also deploy other operations such as attention blocks. Nevertheless, we find that timestep embedding is only injected in the above pipeline with convolutional layers. Based on current standard practice, our study focuses on the pipeline with convolutional layers.

> **Summary of the Problem Statement**
>
> The timestep embedding in NODE and diffusion models is prone to being canceled out due to architectural configuration, especially due to the subsequent normalization layer. We refer to this phenomenon as vanishing timestep embedding. Our claim is that the current architectural configurations for NODE and diffusion models readily cause the vanishing timestep embedding, which should be addressed via architectural correction.

## 3 Solutions

### 3.1 Avoid channel-wise scalar offset: Positional timestep embedding

One perspective on the cause of the vanishing timestep embedding is the addition of the channel-wise scalar offset, which is simply canceled out by the subsequent mean-std normalization. To prevent the vanishing timestep embedding, we introduce a spatial degree of freedom in each channel. Specifically, we propose injecting positional timestep embedding:

$$\mathbf{Z} = \tilde{\mathbf{W}} * \mathbf{X} + \tilde{\mathbf{v}}_t + \tilde{\mathbf{p}}_t. \tag{7}$$

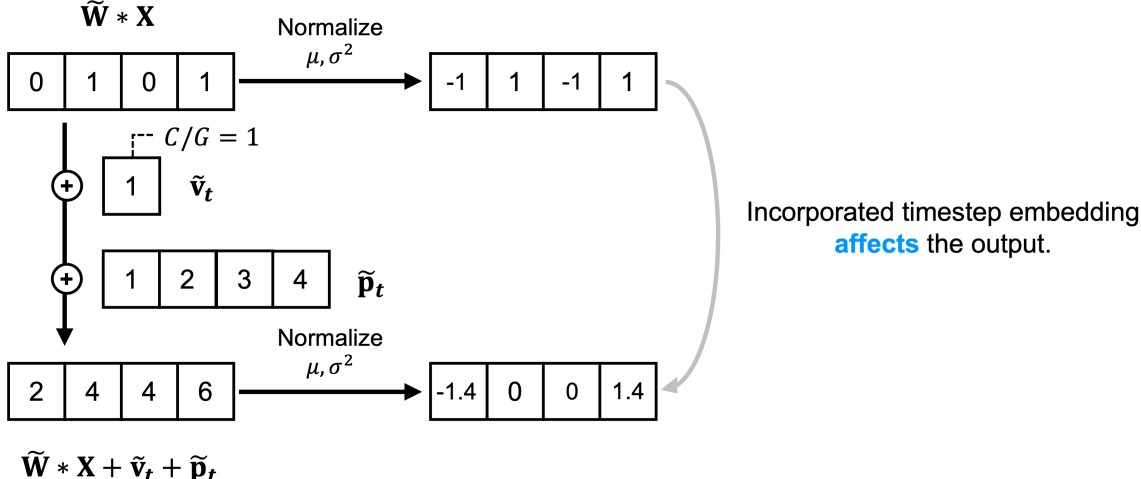

Figure 4: Injecting positional timestep embedding enables a spatial degree of freedom, which is not canceled out by the subsequent normalization

In contrast to the existing timestep embedding $\tilde{\mathbf{v}}_t$ that has a single element in each channel, the positional timestep embedding $\tilde{\mathbf{p}}_t$ has $H \times W$ elements, which are spatially distinct but shared across all channels (Figure 4). Adding this positional timestep embedding enables its different values in a normalization unit, which prevents vanishing timestep embedding, even with channel-wise normalization.

To be compatible with the sinusoidal timestep embedding used in diffusion models, we also propose injecting positional timestep embedding using another parallel branch of MLP from the sinusoidal timestep embedding (Figure 5). Specifically, we generate different frequencies for each position and concatenate their sine and cosine to build the sinusoidal, which is subsequently fed to MLP to produce both the existing timestep embedding and the positional timestep embedding. One may choose a linear-time design of positional timestep embedding $\tilde{\mathbf{p}}_t = t\mathbf{p}$ or an advanced one using MLP with sinusoidal timestep embedding, where we found that both designs are effective in preventing the vanishing timestep embedding.

### 3.2 Control relative variance: Zero bias initialization in convolutions

Another perspective on the cause of the vanishing timestep embedding $\tilde{\mathbf{v}}_t$ is its small variance compared with the variance in the operand, *i.e.*, $\tilde{\mathbf{W}} * \mathbf{X}$. Because GN is applied to the sum of the two, the contribution of the timestep embedding is dependent on their relative scale. If the operand is far more dominant than the timestep embedding, then the contribution of the timestep embedding vanishes. Indeed, Kim et al. (2023) proved that when applying LN to the sum of input $\mathbf{X}$ and operand $\mathbf{P}$ as $\mathrm{LN}(\mathbf{X} + \mathbf{P})$, the gradient $\frac{\partial \mathrm{LN}(\mathbf{X}+\mathbf{P})}{\partial P}$ decays with a factor of $\sqrt{\mathrm{Var}[\mathbf{X}]}$, which means that a larger scale of $\mathbf{X}$ reduces the contribution of $\mathbf{P}$. This theory applies to our analysis: For $\mathrm{GN}(\tilde{\mathbf{W}} * \mathbf{X} + \tilde{\mathbf{v}}_t)$, to encourage alive timestep embedding, we should relatively strengthen the variance of the timestep embedding $\tilde{\mathbf{v}}_t$ and reduce the variance of the operand $\tilde{\mathbf{W}} * \mathbf{X}$.

Indeed, we find that the timestep embedding is inherently prone to vanishing from initialization. Imagine the convolutional kernel $\mathbf{W} = [\mathbf{W}_{1..C}; \mathbf{W}_{C+1}]$ is subjected to an initialization method such as He initialization (He et al., 2015), with zero mean and a variance of $\sigma^2$. Note that convolution computes a summation across elements, including the channel unit and kernel with a spatial size of $n_h \times n_w$. The input feature map $\mathbf{X}$ is the output from the pipeline of [GN–Act], whose variance is $1/2$ at initialization due to the property of the ReLU (He et al., 2015). Thus, the variance of the operand $\tilde{\mathbf{W}} * \mathbf{X}$ is

$$\mathrm{Var}[\mathbf{W}_{1..C} * \mathbf{X}] = \frac{C}{2} n_h n_w \sigma^2. \tag{8}$$

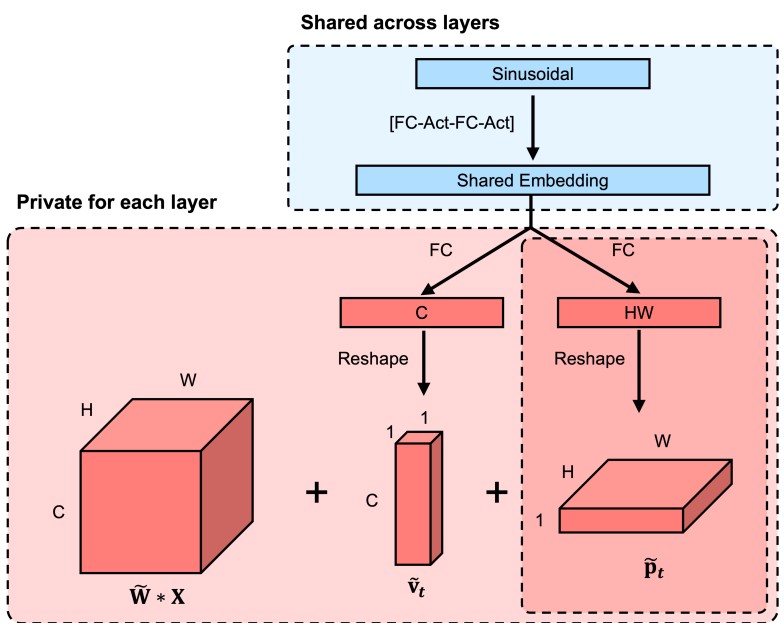

Figure 5: Diffusion models compute sine and cosine from different frequencies and positions, which are fed to MLP to produce timestep embedding $\tilde{\mathbf{v}}_t$. We propose adding another branch to obtain positional timestep embedding $\tilde{\mathbf{p}}_t$ from the sinusoidal.

Table 2: Simulation results to measure empirical ratio for the relative variance

| $C$ | $\mathrm{Var}[\mathbf{W}_{1..C} * \mathbf{X}]$ | $\mathrm{Var}[\mathbf{W}_{C+1} * t\mathbf{J}]$ | Relative Variance |
|---|---|---|---|
| 64 | 0.154 | $4.509 \times 10^{-3}$ | 34.041 |
| 128 | 0.161 | $2.863 \times 10^{-3}$ | 56.089 |
| 256 | 0.166 | $1.399 \times 10^{-3}$ | 118.869 |
| 512 | 0.164 | $6.489 \times 10^{-4}$ | 252.391 |
| 1024 | 0.168 | $3.090 \times 10^{-4}$ | 543.949 |
| 2048 | 0.167 | $1.580 \times 10^{-4}$ | 1056.997 |
| 4096 | 0.166 | $8.223 \times 10^{-5}$ | 2022.146 |

By contrast, the timestep embedding is computed from a single channel, whose summation includes only the kernel unit of $n_h \times n_w$. Thus, we have

$$\mathrm{Var}[\mathbf{W}_{C+1} * t\mathbf{J}] = t^2 n_h n_w \sigma^2. \tag{9}$$

In summary, the relative variance between the operand and the timestep embedding at initialization is

$$\mathrm{Var}[\mathbf{W}_{1..C} * \mathbf{X}]/\mathrm{Var}[\mathbf{W}_{C+1} * t\mathbf{J}] = \frac{C}{2t^2}. \tag{10}$$

In practice, this ratio is quite large because the number of channels $C$ is typically set to a larger number, such as 128. In other words, the timestep embedding inherently exhibits a smaller variance compared with that of the operand. Note that the term $C$ here comes from the input to ConcatConv, not from the input to GN. Specifically, the operand branch computes convolution with an input feature that has $C$ channels; when the number of channels for input feature $C$ is larger, the variance of the operand becomes higher, which affects the relative variance. In other words, the term $C$ here comes with a relative ratio between the operand and the timestep embedding, which is quantified by the ratio of the number of channels at initialization.

We also simulated the ConcatConv module at initialization and measured its empirical variance (Table 2). For this simulation, we set $t = 1$, $n_h = n_w = 3$, $H = W = 32$, $C \in \{64, 128, \cdots, 4096\}$, and measured

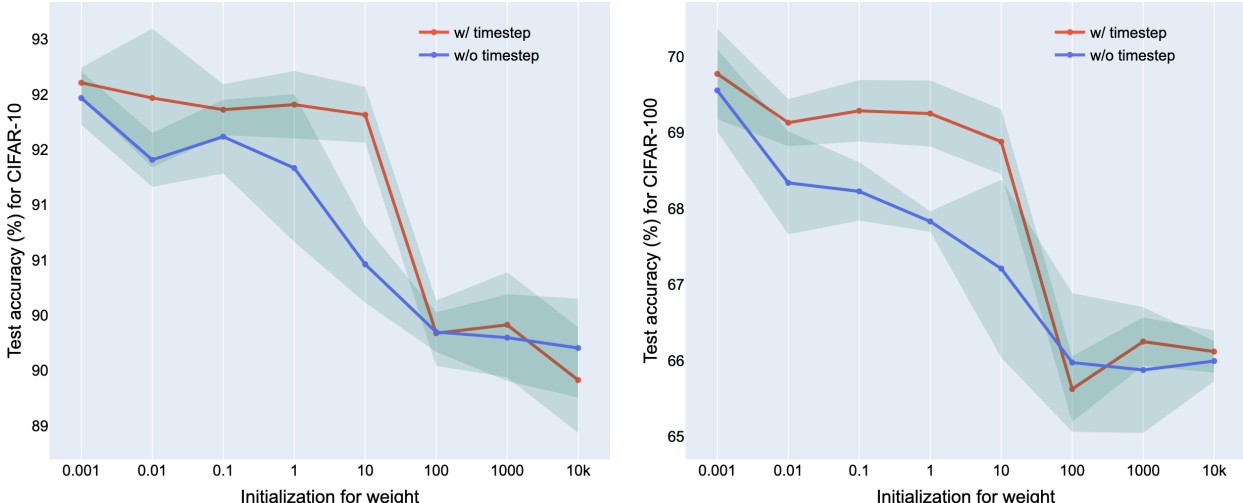

Figure 6: Controlled experiments with different initializations on $\tilde{\mathbf{W}} = \mathbf{W}_{1..C}$ that corresponds to the operand branch with $C$ channels, while keeping $\mathbf{W}_{C+1}$, or equivalently $\mathbf{v}$, that corresponds to the last channel or equivalently timestep branch. The results were obtained from the baseline of the NODE setup using group normalization in Section 4.1, but with different initializations on weights from a standard deviation of $10^{-3}$ to $10^4$. The $x$-axis represents the standard deviation of weight initialization.

variance across 2048 samples. We observed that the measured ratio of relative variance suitably matches with the theoretical expectation of $C/2t^2$, which verifies the above analysis and our claim that timestep embedding is inherently prone to vanish.

Note that we are not saying to increase the variance of the timestep embedding $\tilde{\mathbf{v}}_t$ as much as possible; our claim is that we should at least avoid a dead timestep embedding with few contribution to the output. This objective can be achieved in a variety of ways. For example, increasing the initialization scale in $\mathbf{v}$ and decreasing the initialization scale in $\tilde{\mathbf{W}}$ would be valid. We empirically observed that NODE yielded decreased accuracy in image classification if the initialization scale in $\tilde{\mathbf{W}}$ was larger than $10^2$, which dominates over the timestep embedding (Figure 6). Specifically, the performance of those cases was even similar to that of NODE without timestep embedding, which also exhibited degraded performance due to the effect of weight initialization. Applying a smaller initialization scale in $\tilde{\mathbf{W}}$ leads to alive timestep embedding, which is beneficial to ensure sufficient time-dependency of NODE. Albeit the validity of this approach, however, discovering the proper initialization scale requires hyperparameter tuning depending on the target tasks.

Alternatively, we propose achieving this objective by modifying the initialization of biases. Similar to diffusion models, imagine two branches for the operand via convolution and timestep embedding via MLP, resulting in two distinct bias terms. Note that when there are multiple elements, different bias values within a normalization unit cannot be canceled out by subsequent normalization. In this regard, we propose 1) applying zero bias initialization for convolutions that correspond to the operand, and 2) applying nonzero bias initialization for timestep embedding, allowing default initialization in PyTorch—He initialization (He et al., 2015). Albeit simple, this approach satisfies the objective and works suitably in practice. For the diffusion-style timestep embedding, this approach indicates applying nonzero bias initialization to the MLP. Indeed, the common practice in time-dependent neural networks is to use a consistent initialization for all layers using either zero bias or nonzero bias initialization, whereas we propose adopting different initializations for timestep embedding and operand layers.

### 3.3 Increase degree of freedom in normalization unit: Decrease the number of groups

As discussed above, the channel-wise normalization has a single timestep embedding in each normalization unit, which is simply canceled out. This scenario corresponds to choosing $G = C$ in the GN and is equivalent to IN. When choosing $G = C/2$, each normalization unit has two different timestep embeddings, which would

Table 3: Test accuracy (%) on CIFAR datasets. The standard deviation (Std.) across five runs is presented on the right. $^\dagger$ and $^*$ indicate disobeying and following our guidelines, respectively.

| Architecture | CIFAR-10 | | | CIFAR-100 | | |
|---|---|---|---|---|---|---|
| | Accuracy | Std. | NFE | Accuracy | Std. | NFE |
| Baseline (GN) | 91.76 | (0.24) | 26.0 | 69.47 | (0.54) | 26.0 |
| Replace GN with BN$^\dagger$ | 91.11 | (0.31) | 34.6 | 68.01 | (0.40) | 71.3 |
| + w/o timestep embedding | 91.06 | (0.23) | 28.7 | 68.09 | (0.22) | 69.6 |
| Inject $\tilde{\mathbf{p}}_t$ $^*$ | 91.80 | (0.44) | 26.0 | 69.86 | (0.76) | 26.0 |

not be canceled out but are still prone to vanish owing to the lack of numbers. Choosing a sufficiently smaller $G$ increases the number of different timestep embeddings $C/G$ in each normalization unit, which safeguards against the vanishing timestep embedding. In fact, even though choosing $G = 32$ is the de facto standard, a few implementations of diffusion models such as Score SDE (Song et al., 2021c), ScoreFlow (Song et al., 2021b), and DPM-Solver (Lu et al., 2022) specify $G = \min(C/4, 32)$ in their source code, which restricts the lower bound $C/G \geq 4$. Despite its validity, there has been no explanation for this source code in their paper, nor validation of the lower bound of four. Later, we empirically examine the effect of the number of groups. Indeed, choosing $G = C$ yielded significantly degraded performance, whereas choosing the counterpart in the extreme case of $G = 1$ yielded the most satisfactory results.

## 4 Experiments

We examine the effect of the three proposed methods on solving the vanishing timestep embedding, targeting NODE and diffusion models.

### 4.1 Experiments on NODE

**Model** The target model was NODE for image classification. The early-stage consists of one $3 \times 3$ convolution and two residual blocks with a stride of 2, which produces an initial value for the ODE. The NODE produces output to which a head is applied. The head is a pipeline of [GN–Act–GAP–Dropout–FC–Softmax], where GAP indicates the global average pooling layer. We used ReLU activation function, the number of channels $C = 256$, dropout (Srivastava et al., 2014) with probability of 0.5, and the number of groups $G = 32$ unless specified otherwise. For the ODE, we used relative and absolute tolerances of 0.001 and solver of Runge-Kutta (Runge, 1895; Kutta, 1901) of order 5 of Dormand-Prince-Shampine (dopri5) (Dormand & Prince, 1980) with the adjoint method, implemented in PyTorch (Chen et al., 2018).

**Hyperparamters** The target datasets were CIFAR-{10, 100}, which consist of 60K images of {10, 100} classes (Krizhevsky et al., 2009). For data augmentation, we used $32 \times 32$ random cropping with 4-pixel padding, color jitter with hue degree of 0.05 and saturation degree of 0.05, a random horizontal flip with a probability of 0.5, and mean-std normalization using dataset statistics. For training, the number of epochs of 250, stochastic gradient descent with a momentum of 0.9, learning rate of 0.1, learning rate decay that reduces the learning rate when a metric has stopped improving on a plateau with a patience of 15, weight decay of 0.0001, and mini-batch size of 128 were used. In fact, early studies on NODE used a lower benchmark and reported a test accuracy of 60% for CIFAR-10 (Dupont et al., 2019; Xia et al., 2021), but we targeted a more decent benchmark with a test accuracy above 90% for CIFAR-10, which is similar or even better to the recent results of Kim et al. (2024). An average of five runs with different random seeds was reported. The training was conducted on a single RTX 3090 GPU machine.

**Do not use BN.** We first examine the effect of architectural choice. The baseline performance here was obtained with a NODE-style pipeline with ConcatConv. We observed that NODE yielded decreased accuracy if GN was replaced with BN (Table 3). This observation demonstrates the suboptimal performance of a time-dependent neural network due to the channel-wise normalization of BN. We also report results

Table 4: Test accuracy (%) on CIFAR datasets for different bias initializations

| Zero Bias Initialization | CIFAR-10 | | | CIFAR-100 | | |
|---|---|---|---|---|---|---|
| | Accuracy | Std. | NFE | Accuracy | Std. | NFE |
| Baseline (All) | 92.03 | (0.31) | 26.0 | 69.36 | (0.57) | 26.0 |
| Timestep branch† | 92.00 | (0.33) | 39.2 | 68.91 | (0.79) | 26.0 |
| Convolution branch* | 92.18 | (0.31) | 26.0 | 69.57 | (0.68) | 26.0 |

with BN but without timestep embedding. Indeed, one may think that NODE without timestep embedding would not work suitably, such as with a test accuracy of 10%; nevertheless, we observed that it achieved a suboptimal but quite high test accuracy around 90%. This observation is intriguing because NODE without a timestep embedding is something like a repeating single residual block. We conjecture that the repetition of a single NODE block as well as using the last fully connected layer would be sufficient for basic feature extraction and classification when using rich regularizations such as dropout and weight decay. Furthermore, NODE without timestep embedding can be understood as a fixed-weight RNN with a residual connection, which is already capable of universal approximation, thereby ensuring reasonable performance (Schäfer & Zimmermann, 2006). Note that even when using BN, NODE could exploit other factors such as padding. If padding is applied in ConcatConv, the edges of the timestep embedding exhibit different values, which would not be canceled out by subsequent normalization. However, the edge effect of padding was still ineffective enough to fully solve this problem owing to its suboptimal performance in practice, which was similar to the performance without timestep embedding. This result supports our claim that the vanishing of timestep embedding due to BN is dominant in practice, and other indirect factors, such as padding, affect minor. Furthermore, it significantly increased the number of function evaluations (NFE) due to the ineffectiveness of the edge effect in incorporating a timestep, which required more computational costs. Therefore, we should avoid BN and adopt other proper methods to prevent the vanishing timestep embedding.

**Positional timestep embedding improves performance.** We injected the positional timestep embedding into the baseline above. For both CIFAR-10 and CIFAR-100, the NODE with positional timestep embedding demonstrated improved accuracy compared with the baselines (Table 3). These results were obtained with the same level of NFEs, which indicates that injecting the positional timestep embedding did not harm solving the ODE.

**Apply zero bias initialization to the convolution branch.** We examine the effect of bias initialization. In contrast to previous experiments, the baseline performance here was obtained by a NODE-style pipeline with additive timestep embedding using MLP to deploy two branches with distinct bias terms, where we applied zero bias initialization to both branches of timestep and operand, *i.e.*, convolution. We further obtained results from other setups of applying zero bias initialization to one branch and nonzero bias initialization to another branch. We observed that applying zero bias initialization to the timestep branch slightly degraded performance, whereas applying zero bias initialization to the operand branch improved the test accuracy (Table 4). These results are in agreement with our analysis: To encourage alive timestep embedding, we should apply zero bias initialization to the operand branch.

**Decrease the number of groups.** We examine the choice of the number of groups in GN. Targeting two cases of the number of channels $C = 256$ and $C = 64$, we set the number of groups $G \in \{1, 2, 4, \cdots, C\}$ and evaluated the test accuracy. Figure 7 summarizes the results. As the number of groups increased, the test accuracy dropped. This observation is in agreement with our claim. For example, if $G = C = 256$, the timestep embedding acts as an additive scalar offset, which is canceled out. To avoid this phenomenon, we should ensure a sufficient number of parameters $C/G$ in each group by selecting a smaller number of groups. This analysis applies to the extreme case of $G = 1$, which enables the richest parameters in a normalization unit. Indeed, we observed that $G = 1$ yielded the best performance in most cases.

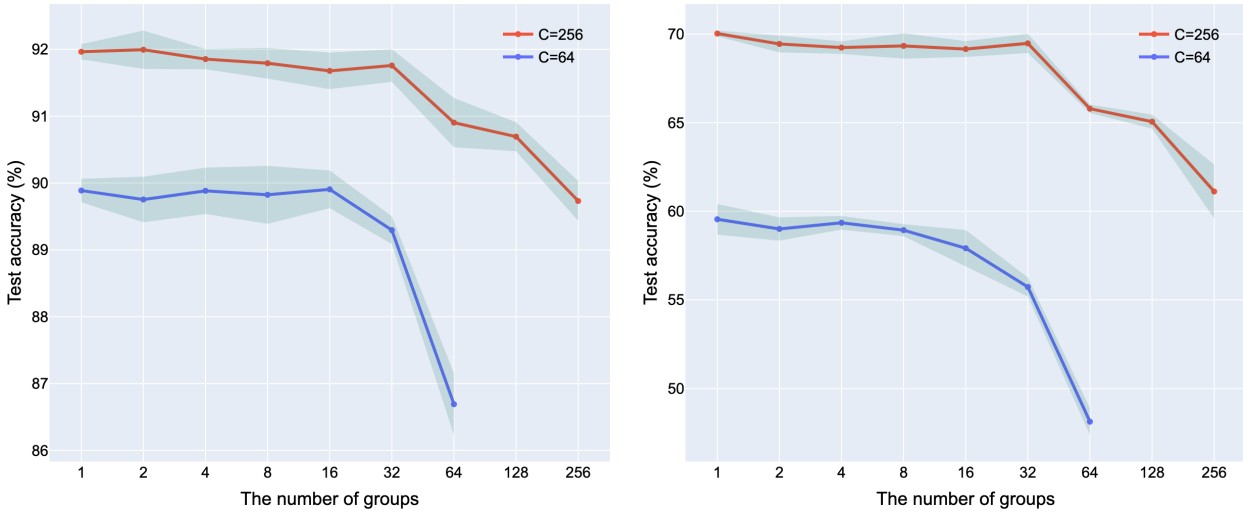

Figure 7: Results for different choices of the number of groups on CIFAR-10 (left) and CIFAR-100 (right). The average and standard deviation from the result of five random seeds are depicted.

Table 5: Results of the DDPM experiment with step-by-step validation of the proposed method

| Setting | Inject $\tilde{\mathbf{p}}_t$ | Zero bias | Set $G = 1$ | FID ($\downarrow$) | IS ($\uparrow$) |
|---|---|---|---|---|---|
| Baseline | ✗ | ✗ | ✗ | 3.238 | 9.507 |
| +Inject $\tilde{\mathbf{p}}_t$ | ✓ | ✗ | ✗ | 3.199 | 9.539 |
| +Zero bias | ✓ | ✓ | ✗ | 3.122 | 9.549 |
| +Set $G = 1$ | ✓ | ✓ | ✓ | 3.074 | 9.603 |

## 4.2 Experiments on Diffusion Models

The target model was DDPM, which is the standard diffusion model for image generation. Because our main target is to confirm the vanishing timestep embedding in existing diffusion models, rather than targeting a custom model, we used the same architecture of existing DDPM. The model uses a U-Net architecture (Ronneberger et al., 2015) with sinusoidal timestep embedding and consists of residual (He et al., 2016) and self-attention blocks (Vaswani et al., 2017). We used Swish activation function (Ramachandran et al., 2018; Elfwing et al., 2018), model width of 128, dropout probability 0.1, and the number of groups $G = 32$ unless specified otherwise. For the diffusion process, $\bar{\beta}_{\max} = 20$ and $\bar{\beta}_{\min} = 0.1$, and noise scales with $\sigma_{\min} = 0.01$ and $\sigma_{\max} = 20$ with a total number of 1000 were used.

The target dataset was CIFAR-10 with $32 \times 32$ resolution, where a random horizontal flip with a probability of 0.5 was applied for data augmentation. For training, the number of iterations of 1300K, Adam optimizer (Kingma & Ba, 2015) without weight decay, learning rate of 0.0002, gradient clipping (Pascanu et al., 2013) with maximum norm of 1, warmup steps of 5K, and mini-batch size of 128 were used. The training was conducted on a $4\times$ RTX 3090 GPU machine, which required approximately seven days for training and sampling.

Table 5 summarizes the results. The three proposed methods improved both the Fréchet inception distance (FID) (Heusel et al., 2017) and the inception score (IS) (Salimans et al., 2016). The improvements were observed through a step-by-step application, which demonstrates the validity of our methods. Note that a common approach to architectural modification, such as stacking more layers, improves FID and inception score but with higher increased computational complexity, such as the number of parameters and FLOPS. However, our proposed methods bring performance gains without largely affecting computational complexity, which we believe is a reasonable improvement. Indeed, the main architecture of U-Net occupies most of the

number of parameters and FLOPS, and the proposed modifications of timestep embeddings affect little on these computational complexities while solving the critical problem that is inherent to the architecture.

## 5 Discussion

**Related works**  To our best knowledge, the vanishing timestep embedding is first discovered by this study. However, existing literature might have unknowingly taken advantage of partially solving this problem. For example, the study on ANODE (Dupont et al., 2019) proposed augmenting an ODE space to improve the representational limitation of NODE. From our analysis, a larger dimension of the ODE increases the number of timestep embedding elements $C/G$ in each normalization unit, which prevents the vanishing timestep embedding. For $G = 32$, when augmenting the dimension of the ODE from 64 to 128, the number of timestep embedding elements changes from two to four, where the latter is more stable against vanishing timestep embedding. However, our theory says that the same effect can be achieved by simply reducing the number of groups from 32 to 16. Furthermore, the larger dimension of ANODE requires a higher computational cost and an increased number of parameters, which is an inefficient approach to solving the vanishing timestep embedding compared with our methods, albeit the validity of augmenting the ODE space to address the representational limitation. Note that the notion of same effect here indicates the equivalence in $C/G$ unit regarding the factor of vanishing timestep embedding, not the equivalence in performance. Figure 7 shows that the models with $C = 256$ achieve improved performance overall compared with that of $C = 64$. In other words, increasing the number of channels $C$ would enlarge the width of the neural network, which itself is advantageous to improve the performance.

In fact, the architecture of vanilla NODE has been tackled by several studies (Zhang et al., 2019; Avelin & Nyström, 2019; Massaroli et al., 2020; Queiruga et al., 2020; Choromanski et al., 2020). They have mentioned that the continuous-time version of residual networks should be incarnated by time-dependency in parameters $\theta$, but the vanilla NODE rather implements time-dependency in $f$ through the design of the neural network. Despite the validity of ensuring time-dependency in parameters $\theta$, the current practice (Dupont et al., 2019; Yan et al., 2020; Norcliffe et al., 2020; Kidger et al., 2021; Xia et al., 2021) still prefers the standard pipeline with ConcatConv owing to its simplicity, and this pipeline has even influenced the implementation of diffusion models. Our study advocates for both theory and practice: The proposed methods are compatible with the existing pipeline of NODE and diffusion models while enriching their insufficient time-dependency.

Also, in the main text, we focused on standard diffusion models, which adopt the DDPM-style pipeline. However, some of the recent diffusion models such as EDM2 (Karras et al., 2024) adopt a modified pipeline, which applies multiplication of the timestep embedding rather than the existing pipeline with the addition of a timestep embedding. Because this modified pipeline does not apply subsequent GN, our analysis says that it is indeed desirable to prevent the vanishing timestep embedding. Nevertheless, we find that their study improved performance without providing a convincing explanation for the underlying cause of the performance improvement. Here, our study enables us to determine whether a timestep embedding would be alive or vanish, which is expected to be a generic criterion for valid architectural design.

**Notes on the known issue of increased NFE**  The dopri5 solver, which is commonly used in NODE, corresponds to solvers with adaptive step size, which choose a large step in smooth dynamics and a small step in varying dynamics. It controls the step size by comparing the predefined tolerance with the current error. For solving NODE, there has been a known issue with increased NFE, which incurs longer computational time. In the official repository of NODE, several users have reported excessively large NFEs in certain usages. The authors of NODE conjectured that the use of the ReLU activation function might yield a stiff ODE due to its discontinuity. This behavior results in a zero step size and an infinite NFE for solving the ODE. They mentioned that when facing this issue, NODE should be modified to use another continuous activation function such as Softplus, which approximates ReLU with a continuous function.

We find that this approach is partially valid but requires correction. Indeed, we empirically observed decreased NFEs when using continuous activation functions such as Softplus and Sigmoid, at the expense of decreased test accuracy (See the Appendix). However, we find that increased NFE can be caused by both stiff and smooth surfaces; when the surface is too smooth such as a plateau, moving a step cannot reach

a desired objective. We observed this phenomenon of increased NFE for NODE with ConcatConv and BN (Table 3), where the timestep embedding indirectly affects the output through the edge effect of padding. To stabilize the ODE, we should ensure that the surface is neither too stiff nor too smooth by regulating an adequate amount of time-dependency.

Therefore, when facing increased NFE, we should confirm whether the problem is caused by large or small stiffness and approach it either by decreasing or increasing stiffness into its stable range. The problem of vanishing timestep embedding corresponds to the latter, and thus our proposed solutions, such as injecting positional timestep embedding, can be a solution to the increased NFE problem.

## 6 Conclusion

This study reported the vulnerability of vanishing timestep embedding in NODE and diffusion models. We claimed that in their architectural pipeline, timestep embedding is prone to vanish when subsequently applying normalization. To address this problem, we proposed 1) adding positional timestep embedding, 2) applying zero bias initialization in the convolution branch, and 3) reducing the number of groups. The proposed methods consistently improved the performance of NODE and diffusion models, which indicates that the two families of models inherently pose this problem, and encouraging alive timestep embedding using these explicit methods is significantly beneficial for their current implementations.

### Acknowledgments

This work was supported by Samsung Electronics Co., Ltd (IO201210-08019-01). Also, this work was supported by JST CREST Grant Number JPMJCR1913.

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

## A  Results on Other Activation Functions

Table 6 summarizes experimental results on other activation functions for NODE. When replacing ReLU with a continuous activation function, we obtained decreased NFE, which leads to faster computation. For the Sigmoid function, we obtained an NFE of 14.0, which is approximately half of the NFE of 26.0 for ReLU. Similar results were confirmed for Softplus, ELU, and SiLU activation functions. Albeit the advantage of decreased NFE, the continuous activation functions yielded decreased test accuracy.

Table 6: Results on other activation functions

| Activation Function | CIFAR-10 | | CIFAR-100 | |
|---|---|---|---|---|
| | Accuracy | NFE | Accuracy | NFE |
| ReLU (Baseline) | 91.76 | 26.0 | 69.47 | 26.0 |
| SiLU | 91.76 | 21.5 | 68.72 | 26.0 |
| ELU | 91.38 | 23.5 | 67.93 | 26.0 |
| Softplus | 89.89 | 18.8 | 65.88 | 16.0 |
| Sigmoid | 89.75 | 14.0 | 65.43 | 14.0 |

## B  Current Implementations of Diffusion Models

The pipeline of [GN–Act–Conv–**Emb**–GN–Act–Conv] has been widely used in current implementations of diffusion models. Here, we provide several examples.

```python
# Reference: https://github.com/hojonathanho/diffusion
def resnet_block(x, *, temb, name, out_ch=None, conv_shortcut=False, dropout):
    # ...
    with tf.variable_scope(name):
        h = x

        h = nonlinearity(normalize(h, temb=temb, name='norm1'))
        h = nn.conv2d(h, name='conv1', num_units=out_ch)

        # add in timestep embedding
        h += nn.dense(nonlinearity(temb), name='temb_proj', num_units=out_ch)[:, None, None,
 :]

        h = nonlinearity(normalize(h, temb=temb, name='norm2'))
        # ...
```

Listing 1: Implementation of DDPM

```python
# Reference: https://github.com/yang-song/score_sde_pytorch
class ResnetBlockDDPM(nn.Module):
    # ...
    def forward(self, x, temb=None):
        # ...
        h = self.act(self.GroupNorm_0(x))
        h = self.Conv_0(h)
        # Add bias to each feature map conditioned on the time embedding
        if temb is not None:
            h += self.Dense_0(self.act(temb))[:, :, None, None]
        h = self.act(self.GroupNorm_1(h))
        # ...
```

Listing 2: Implementation of Score SDE

```python
# Reference: https://github.com/ermongroup/ddim
class ResnetBlock(nn.Module):
  # ...
    def forward(self, x, temb):
        h = x
        h = self.norm1(h)
        h = nonlinearity(h)
        h = self.conv1(h)

        h = h + self.temb_proj(nonlinearity(temb))[:, :, None, None]

        h = self.norm2(h)
        h = nonlinearity(h)
        # ...
```

Listing 3: Implementation of DDIM

```python
# Reference: https://github.com/google-research/vdm
class ResnetBlock(nn.Module):
    # ...
    def __call__(self, x, cond, deterministic: bool, enc=None):
        # ...
        h = x
        h = nonlinearity(normalize1(h))
        h = nn.Conv(
                features=out_ch, kernel_size=(3, 3), strides=(1, 1), name='conv1')(h)

        # add in conditioning
        if cond is not None:
            assert cond.shape[0] == B and len(cond.shape) == 2
            h += nn.Dense(
                    features=out_ch, use_bias=False, kernel_init=nn.initializers.zeros,
                    name='cond_proj')(cond)[:, None, None, :]
```

```
18          h = nonlinearity(normalize2(h))
19          # ...
```

Listing 4: Implementation of VDM

