# OpenReview forum: "Disappearance of Timestep Embedding: A Case Study on Neural ODE and Diffusion Models"
_TMLR — Accepted by TMLR_

### Review · Reviewer_uB6n · 2025-02-05

**Summary Of Contributions:**

The authors argue that the ConcatConv method of embedding time information in convolutional NeuralODEs (NODEs), does not function properly when using GroupNorm (GN) with a large number of groups afterwards. In the extreme case of n_groups = n_channels which is equivalent to using BatchNorm (BN), time information is fully canceled out by the normalization. This can be explained by small (or vanishing) variance of the time embedding contribution relative to the variance of the remaining representation at the normalization step. The issue can also arise in convolutional blocks in diffusion models.

The authors propose that the time embedding will be more prominent when i) using an embedding with positional variance, ii) zeroing biases to reduce variance not related to time information, iii) reducing the number of groups in GN, and show for validation that these modifications may have small positive effects on accuracy of NODEs for image classification on CIFAR-10 /-100 and on image generation in a diffusion model trained on CIFAR-10.

I am not closely familiar with cutting-edge practice in training NeuralODEs and diffusion models, and will therefore not comment on how widespread the use of ConcatConv + GN is and how relevant the contribution is for practitioners.

**Audience:**

Yes

**Broader Impact Concerns:**

/

**Claims And Evidence:**

No

**Requested Changes:**

## 1. Clearer presentation and representation of the results

 - The abstract is vague and nebulous - it should be rewritten to reflect the concrete findings.
 - It would also be much better to not use the term "modern time-dependent networks", which is ill-defined and could as well apply to State-space models, Transformers or RNNs.
 - The title suggests that the timestep information actually disappears in common/practical cases. However this would only be so if BN or equivalently GN with n_groups = n_channels were used in practice, which apparently is not the case.
 - References to current practice for NODE stop at 2021 and for the diffusion models at 2022. Could one or two more recent references be added at least for NODEs?
 - The main motivation given in the beginning is time-series input. But then the experiments are conducted on CIFAR10/100, so static image classification and generation. What is the use of time-information here, how does it relate to augmenting the flow with additional dimensions (ANODE)? The reasons to focus on static tasks instead of time-series input should at least be explained to the reader.
 - (optional) It would be good to extended and improve the discussion of related works. For example, currently no overview of other methods of constructing the time embedding is given, and there is no discussion of the related and much more broadly studied methods of positional encoding in attention blocks. These are only suggestions.
 - Fig.6: Caption and text where Fig.6 is referenced have no details or crossreference to details on what exactly was the model architecture or other parameters here.
 - What is meant by applying nonzero bias init only for the timestep embedding? In my understanding biases in Conv layers are usually added only after summing the convolution over input channels (see e.g. docstring of conv2d in pytorch), so when the time part is already mixed with the operand part, $ W^k \ast [X; tJ] + b^k$ where $k$ is the output channel index. So here there would be no separate biases for the time and the operand parts. Or do the authors in their code only add the time part explicitly after the convolution with $\tilde{W}$ as in eq.4, instead of concatenating $tJ$ before? This could be more clearly stated by adding the biases to the equations.


## 2. Improving the line of arguments and validation experiments

The argumentation from theoretical insight to analysis and solutions is not stringent. Please find my concerns on each part of the argument in the following points.
 - At first, a clear explanation of how BN fully removes the timestep information added by ConcatConv is given. However, then it turns out that BN is never used but GN instead where the problem does typically not arise (and it is not communicated beforehand that the BN discussion is just to introduce to the main topic of GN). Unfortunately, here the theoretical argument breaks off and continues only in anecdotal fashion. The point that ConcatConv results in only a single effective learnable parameter per channel, making the time embedding little expressive, is not expanded upon. The separate issues of small expressivity and of "vanishing" are mixed in the discussion (in sect.3 and also end of sect.2). In sect.2, no arguments are given why the timestep information would still be prone to vanish when GN is used. The concept of relative variances appears only later in the Solutions sect.3.
 - In sect.3 the argument is made that the relative variance of the timestep part may be too small compared to the operand part carrying the sample features. However, the size of these variances is neither analyzed theoretically, nor measured empirically, although both could easily be done (at init for the theory). Nor is any argument given that it is the size of the variance, and not the expressivity of the embedding, which makes the timestep information insufficient. For these reasons, this argument remains in a state of motivation, not of concrete explanation.
 - If small relative variance of the timestep part is detrimental, conversely at the other extreme also small relative operand variance should intuitively become detrimental or result in longer training times. Why is this not observed in Figure 6, where the accuracy remains high also for the smallest W initialization scale? Also, at what point on the x-axis in Fig.6 is the init variance of the operand part equal to that of the time embedding part?
- Fig.6: The drop in performance for large W init scale could well be due to issues with the gradients unrelated to time-embedding. To demonstrate that the effect is due to varying the relative variances of time and input features, a baseline should be shown in the figure where no time embedding was used at all, while the init scale of W is varied as in the experiment.
- The accuracy numbers reported in Tables 2 and 3 are averages across 5 runs. Since the differences in accuracy especially between baseline and the proposed positive intervention is small, the standard error of the mean should be reported (and it might be necessary to reduce it by increasing the number of runs).
- In 4.1, par. "Do not use BN", it is argued that due to padding also with BN some time information enters the network (although with increased NFE), and that this is the reason why the performance would remain relatively high. To make this point properly it would be necessary to provide a baseline where no time embedding is used at all. Would this cause a much larger reduction in accuracy?
- In related work discussion: The authors claim that the ANODE benefit from doubling the internal dimension might be achieved also by halving the number of groups in the GN operation, if the benefit can be explained by making the time-embedding more important. This experiment could simply be performed, and would significantly strengthen the current paper, either by highlighting the importance of the time embedding compared to the dimensionality of the representation, or else by putting the results into a clearer perspective.


Overall, please do not see the number of my concerns as aimed at preventing eventual publication of the paper in TMLR, by possibly requiring the authors to decrease their claims. I strongly believe results only need to be clearly communicated and valid to warrant publication, regardless of hypothetical impact. In this respect also TMLR has no impact requirement, only that some readers would be interested in the results, which I believe will be the case.

**Strengths And Weaknesses:**

## Strengths

- The authors point out a failure mode where timestep embeddings have only a very small contribution to the variance in the normalization layer. If time information is important for the task, such as for complex time-series data, this would be detrimental.
- This may be a problem that has remained under the radar in some off-the-shelf implementations of timestep embedding in convolution based NODEs and diffusion models.
- The optimization of positional encoding has played a significant role in improving transformers on NLP tasks, and analogously optimization of time embedding for models trained on time-series tasks will be important.


## Weaknesses

- Very general claim in the abstract, not adequately supported by the specialized content of the main text, which is limited to ConcatConv + GN for NODEs and scalar time embeddings in conv diffusion models.
- The text focuses on "modern" networks and current practice, but references only up to 2021.
- Strongly limited discussion of related work (e.g. nothing on other methods of time embedding, parallels to positional encoding in attention).
- The validation currently does not establish that the observed improvements in the experiments are due to better time dependency. This is partly because no baseline without time embedding is given, and partly because small differences in accuracy are reported without uncertainty estimates.
- While prominent in the motivation, the paper does not deal with time-series data, but with static tasks (images) where time plays an indirect role as the depth-coordinate in NODEs and the denoising-flow coordinate in denoising models.

In sum, the manuscript exposes a relevant issue in how timestep information is embedded in the standard NODE implementation and some diffusion models. However, these findings are not carefully argued and validated, and overly general claims are made. Therefore I believe significant revisions are required. Please find my detailed concerns below.

---

> ### Author Response · Authors · 2025-03-16
> **Response to Reviewer uB6n (1)**
>
> Thank you for your insightful comments to improve the quality of this manuscript. We find that your comments are valid; in our revised manuscript, we reflected them as much as possible. Here, we provide a point-to-point response to your requested changes.
>
> > 1-1. The abstract is vague and nebulous - it should be rewritten to reflect the concrete findings.
> >
>
> Now we understand that the abstract in the original manuscript was too vague to effectively convey our findings. In the revised abstract, we have explicitly clarified the problem of vanishing timestep embedding due to the current architectural configuration of applying a normalization layer after injecting a timestep embedding. Furthermore, we added several concrete terms, including classification accuracy, FID, and inception score, into the abstract. Thank you for your valuable comment!
>
> > 1-2. It would also be much better to not use the term "modern time-dependent networks", which is ill-defined and could as well apply to State-space models, Transformers or RNNs.
> >
>
> > 1-3. The title suggests that the timestep information actually disappears in common/practical cases. However this would only be so if BN or equivalently GN with n_groups = n_channels were used in practice, which apparently is not the case.
> >
>
> We agree that rewriting the title would improve the clarity of this manuscript. Following your suggestion, we changed the title to “Disappearance of Timestep Embedding: A Case Study on Neural ODE and Diffusion Models” by removing the term of modern time-dependent networks. We also rewrote several sentences in the main text that previously used the term “modern time-dependent networks” into NODE and diffusion models. However, regarding the second comment, we considered several candidates, such as “partial disappearance” or “potential disappearance,” but rather than using these terms, we would like to stick to using the term “Disappearance” for straightforward expression. Indeed, strictly speaking, our claim is that the timestep embedding disappears in common cases due to weight initialization (see our response for comment 2-2) and is not restricted only to the $C/G=1$ case. Nevertheless, we are open to this issue, and if there is further suggestion to improve the clarity of this manuscript, feel free to share your idea with us!
>
> > 1-4. References to current practice for NODE stop at 2021 and for the diffusion models at 2022. Could one or two more recent references be added at least for NODEs?
> >
>
> > 1-6. (optional) It would be good to extended and improve the discussion of related works. For example, currently no overview of other methods of constructing the time embedding is given, and there is no discussion of the related and much more broadly studied methods of positional encoding in attention blocks. These are only suggestions.
> >
>
> Thank you for your suggestion. Following your comment, we checked recent studies on NODE and diffusion models. Firstly, we added a comment on the recent performance of NODE from a study of “Simulation-Free Training of Neural ODEs on Paired Data” (NeurIPS 2024) when referring to their experimental results on CIFAR-10. For diffusion models, we commented on recent architecture by EDM2 (CVPR 2024) in our Discussion section. This architecture injects timestep embedding in a different way, which is worthy of being mentioned in our study. Please check the revised manuscript.
>
> > 1-5. The main motivation given in the beginning is time-series input. But then the experiments are conducted on CIFAR10/100, so static image classification and generation. What is the use of time-information here, how does it relate to augmenting the flow with additional dimensions (ANODE)? The reasons to focus on static tasks instead of time-series input should at least be explained to the reader.
> >
>
> Sorry for any confusion regarding the term “time” in our study. Let us first explain our original intention: In NODE studies, the time unit is something similar to a depth unit in a neural network, not indicating the real notion of time in time-series data. The behavior of a time-dependent function evolving in $t \in [0, T]$ can be understood as a behavior of data passing through layers in a neural network, from the first layer to the last layer, e.g., the 101st layer. In other words, the behavior of a NODE block being time-dependent is similar to a residual network having different functions (strictly speaking, having different weights) across different layers. Thus, repeating a NODE block with respect to different times is similar to passing a residual network across whole layers, which enables performing a static task such as image classification and generation. Similarly, the flow you mentioned propagates with respect to the time unit, much like depth. We hope this explanation convinces you of the exact notion of time in this study. Of course, in our revised manuscript, we clarified this point.

---

> ### Author Response · Authors · 2025-03-16
> **Response to Reviewer uB6n (2)**
>
> > 1-7. Fig.6: Caption and text where Fig.6 is referenced have no details or crossreference to details on what exactly was the model architecture or other parameters here.
> >
>
> Sorry for the inconvenience. We now find that the caption for Fig. 6 lacks sufficient details. The results of Fig. 6 were obtained from our experiments with the baseline of the NODE setup using group normalization in Section 4.1, but with different initializations on weights from a standard deviation of $10^{-3}$ to $10^4$. In our revised manuscript, we have added this clarification to the caption for Fig. 6.
>
> > 1-8. What is meant by applying nonzero bias init only for the timestep embedding? In my understanding biases in Conv layers are usually added only after summing the convolution over input channels (see e.g. docstring of conv2d in pytorch), so when the time part is already mixed with the operand part, $W^k * [X ; tJ] + b^k$ where $k$ is the output channel index. So here there would be no separate biases for the time and the operand parts. Or do the authors in their code only add the time part explicitly after the convolution with $W$  as in eq.4, instead of concatenating $tJ$ before? This could be more clearly stated by adding the biases to the equations.
> >
>
> We apologize that the original explanation regarding the bias is misleading. Again, let us first explain our intention: as the reviewer mentioned, NODE applies ConcatConv, which applies a single bias term to the combination of the operand and the timestep. By contrast, diffusion models apply sinusoidal timestep embedding using MLP, which leads to deploying two separate branches for the operand and timestep embedding, respectively. Specifically, the operand is subjected to convolution with one bias term, whereas timestep embedding is obtained from the MLP with another bias term, resulting in two distinct bias terms. Our bias setup in Section 3.2 implicitly considered this practice, and the results presented in Table 3 were obtained using the two branches. However, we agree that the original explanation was misleading regarding the exact setup. In our revised manuscript, we have clarified the exact scenario of Section 3.2 and explained how the two bias terms come. Again, thank you for spotting this issue; we sincerely appreciate the careful reading and effort of the reviewer to provide constructive feedback.
>
> > 2-1. At first, a clear explanation of how BN fully removes the timestep information added by ConcatConv is given. However, then it turns out that BN is never used but GN instead where the problem does typically not arise (and it is not communicated beforehand that the BN discussion is just to introduce to the main topic of GN). Unfortunately, here the theoretical argument breaks off and continues only in anecdotal fashion. The point that ConcatConv results in only a single effective learnable parameter per channel, making the time embedding little expressive, is not expanded upon. The separate issues of small expressivity and of "vanishing" are mixed in the discussion (in sect.3 and also end of sect.2). In sect.2, no arguments are given why the timestep information would still be prone to vanish when GN is used. The concept of relative variances appears only later in the Solutions sect.3.
> >
>
> Your understanding is correct. Following your first suggestion, we have added an explanation to indicate that the analysis on BN is just an example to introduce before the main topic of GN. Following your second suggestion, in our revised manuscript, we added several sentences to guide readers in understanding why the timestep embedding with GN is still prone to vanishing due to relative variance.

---

> ### Author Response · Authors · 2025-03-16
> **Response to Reviewer uB6n (3)**
>
> > 2-2. In sect.3 the argument is made that the relative variance of the timestep part may be too small compared to the operand part carrying the sample features. However, the size of these variances is neither analyzed theoretically, nor measured empirically, although both could easily be done (at init for the theory). Nor is any argument given that it is the size of the variance, and not the expressivity of the embedding, which makes the timestep information insufficient. For these reasons, this argument remains in a state of motivation, not of concrete explanation.
> >
>
> Thank you for providing a careful checkpoint. After considering your comment, we now present a new theoretical analysis with respect to the relative variance. Note that common weight initialization methods, such as Xavier or He initialization, set weight variance that is inversely proportional to the number of channels. The rationale behind this practice is that the output is computed by summation across all input elements, whose scale is proportional to the number of channels. Here, the operand part would keep a moderate variance due to the summation over channels, whereas the timestep part does not because it has only a single channel. This difference in summation results in a different variance between the two, whose relative ratio is proportional to the number of channels. In other words, our new analysis shows that the timestep branch inherently exhibits a smaller variance compared with that of the operand. Indeed, through empirical measurements, we have observed results that are consistent with this theory, which enables us to quantify the relative variances of the two. If you are interested, please check this new part with theoretical and empirical evidence.
>
> > 2-3. If small relative variance of the timestep part is detrimental, conversely at the other extreme also small relative operand variance should intuitively become detrimental or result in longer training times. Why is this not observed in Figure 6, where the accuracy remains high also for the smallest W initialization scale? Also, at what point on the x-axis in Fig.6 is the init variance of the operand part equal to that of the time embedding part?
> >
>
> This comment is also intriguing. According to our new analysis provided with comment 2-2, we find that the timestep branch inherently exhibits lower variance, up to $C/2$ times. Considering this behavior, it would be advantageous to encourage increased variance for timestep embedding. For the second comment, note that weight initialization is affected by the number of channels, implying that the exact value would differ with layers. In theory, the point where the variance of the operand is equal to that of the timestep embedding would be $C=2t^2 \leq 2$, which would be rarely observed in a practical scenario. We added this point in the new analysis mentioned in response to comment 2-2.
>
> > 2-4. Fig.6: The drop in performance for large W init scale could well be due to issues with the gradients unrelated to time-embedding. To demonstrate that the effect is due to varying the relative variances of time and input features, a baseline should be shown in the figure where no time embedding was used at all, while the init scale of W is varied as in the experiment.
> >
>
> This comment is valid; we should verify whether the different performance arises due to the relative variance or the effect of weight initialization. Following your suggestion, we performed additional experiments, where the baseline performance was obtained without using timestep embedding for each weight initialization case. Indeed, as the reviewer conjectured, the larger initialization itself decreased the performance; nevertheless, we observed that larger initializations tended to exhibit more degradation compared with the setup using timestep embedding. In other words, we observed that both weight initialization and relative variance affected the performance, and considering the overall trend, our claim of controlling relative variance is still valid considering these results. Please check additional results in our revised manuscript.
>
> > 2-5. The accuracy numbers reported in Tables 2 and 3 are averages across 5 runs. Since the differences in accuracy especially between baseline and the proposed positive intervention is small, the standard error of the mean should be reported (and it might be necessary to reduce it by increasing the number of runs).
> >
>
> Thank you for your kind comment. In our revised manuscript, we report the standard deviations of test accuracy in Tables 2 and 3 across five runs. Although the experimental randomness is inevitable in training neural networks, the standard deviations here were quite reasonable to confirm the performance difference. Please check the revised manuscript.

---

> ### Author Response · Authors · 2025-03-16
> **Response to Reviewer uB6n (4)**
>
> > 2-6. In 4.1, par. "Do not use BN", it is argued that due to padding also with BN some time information enters the network (although with increased NFE), and that this is the reason why the performance would remain relatively high. To make this point properly it would be necessary to provide a baseline where no time embedding is used at all. Would this cause a much larger reduction in accuracy?
> >
>
> Based on your suggestion, we performed additional experiments on the setup with BN but without timestep embedding. Indeed, one may think that NODE without timestep embedding would not work suitably, such as with a test accuracy of 10%; nevertheless, we observed that it achieved a suboptimal but quite high test accuracy around 90%. This observation is intriguing because NODE without a timestep embedding is something like a repeating single residual block. We conjecture that the repetition of a single NODE block as well as using the last fully connected layer would be sufficient for basic feature extraction and classification when using rich regularizations such as dropout and weight decay.
>
> Back to the main topic, our objective here was to verify whether timestep embedding would vanish due to BN or would be alive due to padding. According to the aforementioned results, we confirmed that, when using BN, NODE without timestep embedding achieved almost similar performance to that of NODE with timestep embedding. This result supports our claim that the vanishing of timestep embedding due to BN is dominant in practice, and other indirect factors, such as padding, affect minor. Again, thank you for suggesting an insightful comment! The additional results as well as the analysis are attached to the revised manuscript.
>
> > 2-7. In related work discussion: The authors claim that the ANODE benefit from doubling the internal dimension might be achieved also by halving the number of groups in the GN operation, if the benefit can be explained by making the time-embedding more important. This experiment could simply be performed, and would significantly strengthen the current paper, either by highlighting the importance of the time embedding compared to the dimensionality of the representation, or else by putting the results into a clearer perspective.
> >
>
> Thank you for your insightful comment. We find the discussion part was misleading and requires a correction. Firstly, our original intention was to convey that because $C/G$ determines the unit for vanishing timestep embedding, reducing the number of groups $G$ would be equivalent to increasing the number of channels $C$. However, the equivalence means the equivalence in unit $C/G$, i.e., the factor of vanishing timestep embedding, not the equivalence of performance of NODE. Indeed, increasing the number of channels $C$ would enlarge the width of the neural network, which itself is advantageous to improve the performance. As evidence, Figure 7 shows that the models with $C=256$ achieve improved performance overall compared with that of $C=64$. In other words, the performance of NODE is affected by both the factor of vanishing timestep embedding and width determined by the number of channels $C$. In our revised manuscript, we clarified that achieving the same effect by increasing the number of channels $C$ indicates the equivalence in $C/G$ unit regarding the factor of vanishing timestep embedding, not the equivalence in performance.
>
> Finally, please check our revised manuscript, where changed or added parts are colored with blue. We have learned a lot from your suggestions and feel that your comments further improved our manuscript. Thank you once again for providing kind comments!

---

> > ### Comment · Reviewer_uB6n · 2025-03-29
> > **Response to revision**
> >
> > I would like to thank the authors very much for their work on the revision and the detailed response.
> > While I still think that significant issues with the presentation remain, the revised manuscript has clearly improved the experimental and theoretical analysis.
> > Below, I list a few points for minor revisions which I would recommend to further improve the final manuscript. Nonetheless, I think the manuscript now provides better and clearer evidence, and I am leaning to recommend acceptance to the editor.
> >
> > ### Remaining minor issues
> > - I appreciate the changes to the title and abstract. I understand the argument to keep the "disappearance of time-step embedding" formulation in the title. However, the abstract itself could still be made much more objective and informative about the actual content of the analysis. It does not mention the three crucial terms with play the main role in the manuscript: Concatconv, BatchNorm, and GroupNorm. The abstract would be much improved if it summarized the results with reference to these ingredients.
> > - Time and depth in NODE: Maybe it would be good to point out that time-series data *could* be used, but in the present manuscript only static tasks with a single input at t=0 and output at t=T are used. Also a motivation for this choice could be given.
> > - Fig.6 and sect.3.2: Thank you for adding the baseline without time-step and the very useful theoretical calculation of the variance.
> >    * Shouldn't the number of channels in one group, C/G, be the variance one needs to compare to instead of the total number of channels C ?
> >    * Currently the description in caption and in sect.3.2 of the experiment in Fig.6 was unclear to me: The caption says the standard GroupNorm framework was used while varying the initialization of the weights. However, according to the calculation in sect3.2 this would mean the relative variance of operand and timestep embedding does not change since $\sigma$ enters in both of them. Instead the description in the text on page 9 claims to vary only the initialization of $\tilde{W}$ while keeping the same $\tilde{v}$, so that only the variance of the operand part changes, which would make sense. Please adapt caption and text to clarify which of the two experiments was done.
> > - That NODE without timestep embedding still achieves relatively good performance is expected because it corresponds to a recurrent network with fixed weights. Such models are capable of universal approximation (see e.g. Schaefer \& Zimmermann 2006, doi: 10.1007/11840817_66 ) if the recurrent block is wide enough compared to the input width, and can also represent time implicitly in their dynamical state if needed (which is not the case here since the tasks are static). Adding timestep information explicitly may then add a representation of time that may be more suitable for the task.

---

> > > ### Author Response · Authors · 2025-03-31
> > > **Response to Reviewer uB6n for Minor Issues (1)**
> > >
> > > Thank you for hearing from you again and for your true efforts to improve our manuscript! We find that your new comments are actionable immediately, so we respond right here.
> > >
> > > > I appreciate the changes to the title and abstract. I understand the argument to keep the "disappearance of time-step embedding" formulation in the title. However, the abstract itself could still be made much more objective and informative about the actual content of the analysis. It does not mention the three crucial terms with play the main role in the manuscript: Concatconv, BatchNorm, and GroupNorm. The abstract would be much improved if it summarized the results with reference to these ingredients.
> > > >
> > >
> > > We agree with your point. We now understand that the abstract lacked the three terms. Indeed, a brief summary of our core findings is worthy of being mentioned in the abstract. In our revised abstract, we wrote, “Specifically, we find that the ConcatConv operation, which is widely used in neural ordinary differential equations, causes an additive effect of timestep embedding, which is readily canceled out by the subsequent batch normalization. This vanishing timestep embedding also arises for group normalization and is analyzed thoroughly with respect to the number of channels, groups, and relative variance.” Now we feel that the abstract became further high quality. Thank you again for your kind comment!
> > >
> > > > Shouldn't the number of channels in one group, C/G, be the variance one needs to compare to instead of the total number of channels C ?
> > > >
> > >
> > > Thank you for your careful reading of the new part. As you mentioned, our claim regarding GroupNorm considers the $C/G$ unit when considering the input to GroupNorm. By contrast, the term $C$ in the new analysis part comes from the input to ConcatConv, not from the input to GroupNorm. Specifically, the operand branch computes convolution with an input feature that has $C$ channels; when the number of channels for input feature $C$ is larger, the variance of the operand becomes higher, which affects the relative variance. In other words, the term $C$ here comes with a relative ratio between the operand and the timestep embedding, which is quantified by the ratio of the number of channels at initialization. We hope that this clarification will convince you that the new analysis would require the term $C$ as the number of channels in the input, which affects relative variance. Nevertheless, we think that this part requires further clarification, and we have updated this point. Thank you for your valuable comment!

---

> > > ### Author Response · Authors · 2025-03-31
> > > **Response to Reviewer uB6n for Minor Issues (2)**
> > >
> > > > Currently the description in caption and in sect.3.2 of the experiment in Fig.6 was unclear to me: The caption says the standard GroupNorm framework was used while varying the initialization of the weights. However, according to the calculation in sect3.2 this would mean the relative variance of operand and timestep embedding does not change since $\sigma$ enters in both of them. Instead the description in the text on page 9 claims to vary only the initialization of $W$ while keeping the same $v$, so that only the variance of the operand part changes, which would make sense. Please adapt caption and text to clarify which of the two experiments was done.
> > > >
> > >
> > > Sorry for the confusion. Indeed, our original intention corresponds to the latter in your comment. In the experiments presented in Fig. 6, we keep the variance of $W_{C+1}$, or equivalently $v$. However, we apply controlled initialization to $W_{1..C}$. In other words, the initialization applies to $C$ channels while leaving the last channel. We now find that this point should be clarified more exactly. Thank you once again for providing careful checkpoints, and please see the revised manuscript where this point has been clarified.
> > >
> > > > Time and depth in NODE: Maybe it would be good to point out that time-series data could be used, but in the present manuscript only static tasks with a single input at t=0 and output at t=T are used. Also a motivation for this choice could be given.
> > > >
> > >
> > > > That NODE without timestep embedding still achieves relatively good performance is expected because it corresponds to a recurrent network with fixed weights. Such models are capable of universal approximation (see e.g. Schaefer & Zimmermann 2006, doi: 10.1007/11840817_66 ) if the recurrent block is wide enough compared to the input width, and can also represent time implicitly in their dynamical state if needed (which is not the case here since the tasks are static). Adding timestep information explicitly may then add a representation of time that may be more suitable for the task.
> > > >
> > >
> > > Thank you for your professional comments. Please let us respond to the two comments together. Indeed, the relatively good performance of NODE without timestep embedding is intriguing, and its explanation is worthy of being expanded. Your suggestion of understanding this model as a fixed-weight RNN is also valid when the RNN has a residual connection, which would support understanding this observation. We have added this RNN case as a motivational example in Section 2.1 and also mentioned that time-series data could be used. In Section 4.1, we have added the suggested reference to better understand NODE without timestep embedding as a fixed-weight RNN. If you are interested, please check the revised manuscript.
> > >
> > > Finally, please check our revised manuscript, where changed or added parts are colored. Note that the coloring is applied based on the differences between this revised version at 31 March and the initial submission, not with the recent revised version at 17 March. Thank you again for the affirmative comment!

---

> > > > ### Comment · Reviewer_uB6n · 2025-03-31
> > > >
> > > > Thank you, I will recommend acceptance.

---

### Review · Reviewer_74vv · 2025-02-08

**Summary Of Contributions:**

This paper explores the impact of architectural choices in time-dependent neural networks on their ability to retain time-awareness. The authors argue that the design of time step embeddings significantly affects model performance, particularly in diffusion models.

**Audience:**

Yes

**Claims And Evidence:**

Yes

**Requested Changes:**

- Clearly Define the Vanishing Time Step Problem: The authors should explicitly state what constitutes the disappearance of time step embedding and why it is a problem. Providing mathematical or empirical evidence would be beneficial.

- Demonstrate the Utility of Non-Vanishing Time Step Embeddings: Instead of assuming its usefulness, the authors should first provide a compelling demonstration that maintaining time step information is desirable in practical applications.

- Establish a Concrete Link Between Time Step and Performance: The paper should clarify how time-awareness (or its absence) affects final performance in image modeling tasks like diffusion models. The relationship should be explicitly demonstrated rather than assumed.

**Strengths And Weaknesses:**

Strengths
- Relevant Topic: The study addresses an important aspect of time-dependent neural networks, which has implications for various applications, including time series modeling and diffusion models.

Weaknesses
- Unclear Problem Definition: The vanishing time step problem is not clearly defined. The paper lacks a precise formulation of the issue, making it difficult to assess its significance.
- Weak Justification of Non-Vanishing Time Step: The connection between non-vanishing time step embeddings and final model performance remains vague. It is not convincingly demonstrated whether ensuring non-vanishing time step embeddings is beneficial.

---

> ### Author Response · Authors · 2025-03-16
> **Response to Reviewer 74vv**
>
> Thank you for your careful comments to improve the quality of this manuscript. We find that your comments are valid; in our revised manuscript, we reflected them as much as possible. Here, we provide a point-to-point response to your requested changes.
>
> > Clearly Define the Vanishing Time Step Problem: The authors should explicitly state what constitutes the disappearance of time step embedding and why it is a problem. Providing mathematical or empirical evidence would be beneficial.
> >
>
> Thank you for your valuable comment. We now understand that the problem statement in the original manuscript was quite verbal and may have been difficult to spot in the main text. Following your suggestion, we have explicitly written the problem statement in a colored box in the revised manuscript. We believe that this part would enable readers grasp our core finding at a glance.
>
> Furthermore, in our revised manuscript, we also strengthened the theoretical and empirical evidence for our claim. Specifically, the smaller contribution of timestep embedding with respect to the relative variance was further analyzed, where we show that its contribution is inherently smaller due to weight initialization. This result is also verified by empirical measurements, whose results were consistent with our theoretical analysis. If you are interested, please see our revised manuscript.
>
> > Demonstrate the Utility of Non-Vanishing Time Step Embeddings: Instead of assuming its usefulness, the authors should first provide a compelling demonstration that maintaining time step information is desirable in practical applications.
> >
>
> > Establish a Concrete Link Between Time Step and Performance: The paper should clarify how time-awareness (or its absence) affects final performance in image modeling tasks like diffusion models. The relationship should be explicitly demonstrated rather than assumed.
> >
>
> Thank you for the careful comments; please let us answer the two comments together. We agree that providing more convincing results regarding the vanishing timestep embedding would improve the quality of our study. In our revised manuscript, we attached additional experimental results where NODE adopts BN but without the timestep embedding. We compared the performance of this case with the setup of using both BN and timestep embedding, where we observed that the two exhibited similar performance. This result implies that the vanishing timestep embedding dominates for NODE with BN, and other indirect factors, such as padding, contribute little, validating our claim. We hope that this new part would convince you of the clear and strict validation of the vanishing timestep embedding.
>
> Finally, please check our revised manuscript, where changed or added parts are colored with blue. We have learned a lot from your suggestions and feel that your comments further improved our manuscript. Thank you once again for providing valuable comments!

---

### Review · Reviewer_xR1m · 2025-03-04

**Summary Of Contributions:**

This work studies the disappearance of tilmestep embeddings in neural networks that model a time-varying function, example neural ODEs and diffusion models. This paper analyzes few neural networks with timestep input and shows that some architecture choices (such as batch-normalization) lead to vanishing time-step embeddings, resulting in loss of tilmestep-aware modeling capabilities.  It also proposes a solution that includes incorporating positional embeddings after the tilmestep embedding to negate the effect of batch-normalization layers. Numerical experiments (for Neural ODEs and Diffusion Models) are performed to strengthen these arguments in NODE and diffusion model training.

**Audience:**

Yes

**Claims And Evidence:**

No

**Requested Changes:**

I believe answering the following questions would help strengthen this work.
1. Diffusion models also incorporate tilmestep embeddings through modulation layers as well, which gets injected into each block in the form of a gating mechanism. How does timestep embedding vanishes in such instances?
2. How does one extend this analysis to transformer networks modeling time-dependent function?
3. In Tab.2, even the solution with batch-norm achieves close to the optimal performance. Doesn’t it signify that tilmestep embeddings do not vanish to the extend this work shows it does?
4. In Tab.4, why is it the case that the baseline performance is very close to the performance of the scheme with all mitigations applied (for instance the inception scores are nearly similar)?
5. For diffusion models, have you tried to explore the effects of timestep disappearance with more depth in the network? Does it worsen this issue?

**Strengths And Weaknesses:**

Strengths:
1. Simple solutions to address the issue of vanishing tilmestep embedding influence in a neural network modeling a time-varying function.
2. Helps provide a simple justification to the use of GroupNorm over BatchNorm as well as C/G factor in diffusion models.

Weaknesses:
1. It’s unclear if in practice, such tilmestep embedding vanishes to such an extent that the model is unable to learn the time-dependent behavior. For instance, in Tab.2, even with batch-norm, it was able to recover very close to the optimal performance. Similarly, In Tab. 4, there’s not a lot of difference between the baseline and the scheme with all three mitigations in terms of the inception score.
2. Although this work performs experiments on Diffusion models which injects timestep embedding through convolutional layers, many recent and advanced architectures injects the timestep embedding through modulation layers. It’s unclear how the proposed claims/arguments apply to these diffusion models.

---

> ### Author Response · Authors · 2025-03-16
> **Response to Reviewer xR1m (1)**
>
> Thank you for your valuable comments to improve the quality of this manuscript. We find that your comments are valid; in our revised manuscript, we reflected them as much as possible. Here, we provide a point-to-point response to your requested changes.
>
> > Diffusion models also incorporate tilmestep embeddings through modulation layers as well, which gets injected into each block in the form of a gating mechanism. How does timestep embedding vanishes in such instances?
> >
>
> Thank you for the professional comment on diffusion models. Firstly, in the original manuscript, Section 2.2 explained how the timestep embedding vanishes for both NODE and diffusion models. We found that common diffusion models adopt the DDPM-style pipeline of [GN-Act-Conv-Emb-GN-Act-Conv], which uses sinusoidals with MLP but is prone to vanishing due to the subsequent GN. In other words, our main analysis seamlessly applies to these diffusion models. However, as you mentioned, some of the recent diffusion models such as EDM2 (CVPR 2024) adopt a modified pipeline, which applies multiplication of the timestep embedding rather than the existing pipeline with the addition of a timestep embedding. Because this modified pipeline does not apply subsequent GN, our analysis says that it is indeed desirable to prevent vanishing timestep embedding. Nevertheless, we find that their study improved performance without a convincing explanation of the underlying cause of the performance improvement. Here, our study enables us to determine whether a timestep embedding would be alive or vanish, which is expected to be a generic criterion for valid architectural design. Overall, we find that this discourse is worthy of being mentioned. In our revised manuscript, we mentioned the pipeline you suggested in the discussion section. Thank you again for spotting this issue.
>
> > How does one extend this analysis to transformer networks modeling time-dependent function?
> >
>
> This is an intriguing topic. Although our research focuses on the vulnerability of the common practice of injecting time-dependency, such as ConcatConv and sinusoidal timestep embedding, it can be extended to other transformer architectures. When one tries to design their own architecture of a transformer that incorporates timestep embedding, being aware of the behavior of the normalization layer in relation to timestep embedding would enable them to obtain a valid architecture and achieve higher performance. For example, one can apply our three guidelines to prevent vanishing timestep embedding for their custom transformers. Note that the current design of time-dependent neural networks, such as NODE, still prefers convolutional neural networks; transformers or MHSA are rarely used in the design of NODE. Nevertheless, this direction of using transformers for modeling time-dependent functions is likely to be discussed in follow-up studies, where our research is expected to be useful material.
>
> Also, just in case, please let us clarify that the term “time” in our study differs from the conventional notion of time used in time-series data. In NODE studies, the time unit is something similar to a depth unit in a neural network, not indicating the real notion of time for time-series data. The behavior of a time-dependent function evolving in $t \in [0, T]$ can be understood as a behavior of data passing through the layers of a neural network, from the first layer to the last layer, e.g., the 101st layer. In other words, the behavior of a NODE block being time-dependent is similar to a residual network having different functions or weights across different layers. Thus, repeating a NODE block with respect to different times is similar to passing a residual network across whole layers, which enables performing a static task such as image classification and generation. We hope this explanation convinces you of the exact notion of time in this study. Of course, in our revised manuscript, we clarified this point.

---

> ### Author Response · Authors · 2025-03-16
> **Response to Reviewer xR1m (2)**
>
> > In Tab.2, even the solution with batch-norm achieves close to the optimal performance. Doesn’t it signify that tilmestep embeddings do not vanish to the extend this work shows it does?
> >
>
> Based on your suggestion, we performed additional experiments on the setup with BN but without timestep embedding. Indeed, one may think that NODE without timestep embedding would not work suitably, such as with a test accuracy of 10%; nevertheless, we observed that it achieved a suboptimal but quite high test accuracy around 90%. This observation is intriguing because NODE without a timestep embedding is something like a repeating single residual block. We conjecture that the repetition of a single NODE block as well as using the last fully connected layer would be sufficient for basic feature extraction and classification when using rich regularizations such as dropout and weight decay.
>
> Back to the main topic, our objective here was to verify whether timestep embedding would vanish due to BN or would be alive due to padding. According to the aforementioned results, we confirmed that, when using BN, NODE without timestep embedding achieved almost similar performance to that of NODE with timestep embedding. This result supports our claim that the vanishing of timestep embedding due to BN is dominant in practice, and other indirect factors, such as padding, affect minor. Again, thank you for suggesting an insightful comment! The additional results as well as the analysis are attached to the revised manuscript.
>
> > In Tab.4, why is it the case that the baseline performance is very close to the performance of the scheme with all mitigations applied (for instance the inception scores are nearly similar)?
> >
>
> This is a valid comment. A common approach to architectural modification, such as stacking more layers, improves FID and inception score but with higher increased computational complexity, such as the number of parameters and FLOPS. However, our proposed methods bring performance gains without largely affecting computational complexity, which we believe is a reasonable improvement. Indeed, the main architecture of U-Net occupies most of the number of parameters and FLOPS, and the proposed modifications of timestep embeddings affect little on these computational complexities while solving the critical problem that is inherent to the architecture. In the revised manuscript, we added this explanation to clarify the meaning of the performance gain.
>
> > For diffusion models, have you tried to explore the effects of timestep disappearance with more depth in the network? Does it worsen this issue?
> >
>
> Thank you for your careful reading. First of all, please understand that our main target in the original manuscript was to confirm the vanishing timestep embedding in existing diffusion models, rather than targeting a custom model. Thus, we used the same architecture of existing DDPM and confirmed that the timestep embedding vanished in commonly used diffusion models. Essentially, the phenomenon of vanishing timestep embedding arises from the architectural configuration when injecting sinusoidal timestep embedding into diffusion models, which is adopted across whole residual blocks in U-Net. Therefore, our analysis says that the U-Net with more depth would have the same problem unless the underlying architectural problem is not addressed. For this point, in the revised manuscript, we clarified the choice of the target model in the experiments section.
>
> Finally, please check our revised manuscript, where changed or added parts are colored with blue. We have learned a lot from your suggestions and feel that your comments further improved our manuscript. Thank you once again for providing careful checkpoints!

---

### Review · Reviewer_TPBA · 2025-03-12

**Summary Of Contributions:**

This paper proposes methods for solving the vanishing timestep embedding in ordinary neural differential equations.

**Audience:**

Yes

**Claims And Evidence:**

Yes

**Requested Changes:**

See the comments in strengths and weaknesses.

**Strengths And Weaknesses:**

The three methods proposed in this paper, including positional timestep embedding, zero bias initialization, and fewer groups, are intuitive and simple, while the performance improvement is not significant. I would be more excited if the author could discuss the solutions with more depth in theory and in a broader scope. For example, how NODE can maintain the time information with heterogeneity of neurons' time constants which can approximate the Laplacian basis.

The paper overall is well-written and contains sufficient details. I don't have specific criticism on techniques.

---

> ### Author Response · Authors · 2025-03-16
> **Response to Reviewer TPBA**
>
> Thank you for your thoughtful comments to improve the quality of this manuscript. We find that your comments are valid; in our revised manuscript, we reflected them as much as possible. Here, we provide a point-to-point response to your requested changes.
>
> > The three methods proposed in this paper, including positional timestep embedding, zero bias initialization, and fewer groups, are intuitive and simple, while the performance improvement is not significant. I would be more excited if the author could discuss the solutions with more depth in theory and in a broader scope. For example, how NODE can maintain the time information with heterogeneity of neurons' time constants which can approximate the Laplacian basis.
> >
>
> This is a valid comment. Firstly, note that a common approach to architectural modification, such as stacking more layers, improves FID and inception score but with higher increased computational complexity, such as the number of parameters and FLOPS. However, our proposed methods bring performance gains without largely affecting computational complexity, which we believe is a reasonable improvement. Indeed, the main architecture of U-Net occupies most of the number of parameters and FLOPS, and the proposed modifications of timestep embeddings affect little on these computational complexities while solving the critical problem that is inherent to the architecture. In the revised manuscript, we added this explanation to clarify the meaning of the performance gain.
>
> Secondly, as the reviewer mentioned, to extend our analysis in a broader scope, we added new theoretical and empirical evidence regarding relative variance. Specifically, the new analysis shows that timestep embedding inherently contributes a smaller amount compared with the operand. This behavior can be easily observed in general initialization methods, such as Xavier or He initialization. This analysis also supports our claim of vanishing timestep embedding due to architectural configuration. If you are interested, see our revised manuscript.
>
> > The paper overall is well-written and contains sufficient details. I don't have specific criticism on techniques.
> >
>
> Thank you for the affirmative comment. Note that to strengthen and support our claim, we also attached additional experimental results in the revised manuscript. Specifically, we performed strict validation to verify the effect of weight initialization and relative variance by comparing the original results and baseline performance in the setup without a timestep embedding.
>
> Finally, please check our revised manuscript, where changed or added parts are colored with blue. We have learned a lot from your suggestions and feel that your comments further improved our manuscript. Thank you once again for providing thoughtful checkpoints!

---

### Decision · Action_Editor_g8i5 · 2025-05-19

**Recommendation:** Accept as is

**Comment:**

All reviewers but one lean towards acceptance of the paper.

The consensus among reviewers is that the issues underlined by this paper are real, but they might not be so severe in practice due to the discrepancy between the specific layers studied in this work and the layers used by practitioners. Furthermore, the fact that the improvement observed in the experiments is due to the fixation of the issue of disappearance ot time embedding could be clearer.

Still, all reviewers found the message of this work interesting, that it will interest several researchers and maybe pave the way to a better understanding of timestep embeddings in recent neural networks.

**Audience:**

Many ML researchers work with such architectures, so it is interesting for them the problems mentioned by this paper are underlined and fixed.

**Claims And Evidence:**

The claims of this paper are:
- The standard way to make layers time dependent in neural odes / diffusion models, using concatconv, has a trivial dependency on time when paired with group-norm layers, as is usually the case. This claim is well supported both by hand computations and by numerical experiments
- Incorporating positional information allows to fix this issue, and yields models that are truly time-dependent. This claim is also well supported by numerical evidence, and the intuition behind it is clear in the text.

---

> ### Author Response · Authors · 2025-05-22
> **Thank you for your kind services.**
>
> Dear Action Editor,
>
> Thank you for the affirmative decision and evaluation of the findings of this study.
>
> We have uploaded the deanonymized, camera-ready version of this manuscript. We confirm that the camera-ready manuscript is consistent with the TMLR format. We have also double-checked the author information and included the OpenReview URL in this version.
>
> Thank you once again for your kind services!
>
> Best regards,
>
> Paper 3920 Authors.